

# 1 Aerosol optical characteristics in the urban area of Rome, Italy, and their impact
# 2 on the UV index.

Monica Campanelli[1], Anna Maria Siani[2], Alcide di Sarra[3], Anna Maria Iannarelli[4], Paolo Sanò[1], Henri
Diémoz[5], Giampietro. Casasanta[1], Marco Cacciani[2], Luca Tofful[6], Stefano Dietrich[2]
[1] Institute of Atmospheric Sciences and Climate, National Research Council, Rome, Italy
[2] Sapienza University of Rome, Department of Physics, Rome, Italy
[3] Dipartimento Ambiente, Cambiamenti Globali e Sviluppo Sostenibile, Ente per le Nuove Tecnologie,
l'Energia e l'Ambiente, Rome, Italy
[4] SERCO, Italy
[5] Agenzia Regionale Protezione Ambientale-Valle d'Aosta, ARPA-VDA, Italy
[6] Institute of Atmospheric pollution, National Research Council, Italy
**Abstract**
The aerosol optical characteristics in the urban area of Rome were retrieved over a period of 7 years
from March to September 2010-2016. The impact of aerosol single scattering albedo (SSA), optical
depth (AOD), estimated at 400 nm, and Ångström exponent on the ultraviolet (UV) index has been
analyzed. Aerosol optical properties are provided by a PREDE-POM sun-sky radiometer of the
ESR/SKYNET network and the UV index values were retrieved by a Brewer spectrophotometer both
located in Rome. Chemical characterization of urban PM10 (particulate matter 10 micrometers or less
in diameter) samples, collected during the URBan Sustainability Related to Observed and Monitored
Aerosol (URBS ROMA) intensive filed campaign held in summer 2011 in the same site, was performed.
PM macro-components were grouped in order to evaluate the contribution of the main macro-sources
(SOIL, SEA, SECONDARY INORGANIC, ORGANICS and TRAFFIC) and the analysis of the
modulation of their concentration was found to strongly affects the absorption capability of the
atmosphere over Rome. The surface forcing efficiency, provided by the decreasing trend of UV index
with AOD, which is the primary parameter affecting the surface irradiance, was found very significant,
probably masking the dependence of UV index on SSA and Ångström exponents. Moreover it was found
greater for larger particles and with a more pronounced slope at the smaller solar zenith angle. In Rome
large particles are generally less absorbing since related to the presence of SOIL and SEA components
in the atmosphere. The former contribution was found much higher in summer months because of the
numerous episodes of Saharan dust transport



## 1. Introduction

The aerosol influence on the incoming and outgoing solar radiation is a widely studied topic because of its relation with the Earth's radiative balance and climate. The aerosol influence on ultraviolet (UV) solar irradiance is also very important, particularly in urban areas, nevertheless still uncertain. In fact, the aerosol capability of absorbing UV radiation has important implications for tropospheric photochemistry, human health, and agricultural productivity (Dickerson et al., 1997; He and Carmichael, 1999; Castro et al., 2001; Casasanta et al., 2011; Mok et al. 2018).

The aerosol single scattering albedo (SSA), that is the ratio of the aerosol scattering to extinction coefficient, representing an index of the aerosol absorption capability, and the optical depth (AOD), are important radiative parameters to determine the aerosol effect on the UV irradiance at the surface.

Reuder and Schwander (1999) demonstrated that more than 80% of the aerosol effect on surface UV radiation due to increasing turbidity of the atmosphere can be estimated through aerosol optical depth and single scattering albedo.

UV absorption by aerosol, characterized by low SSA values at wavelengths shorter than 400 nm, is commonly attributed to organic aerosols that absorb predominantly in the UV region and show a stronger wavelength dependence than a purely black carbon absorption (Kirchstetter et al., 2004). Also mineral components shows a significant absorption in the UV region, as highlighted by Meloni et al. (2006).

Martins et al. (2009) indicated that the absorption efficiency of urban aerosol is considerably larger in the UV than in the visible and is probably linked to the absorption by organic aerosol. Similarly, an enhancement of aerosol absorption at UV wavelengths was observed in urban cities such as Rome, Italy (Ialongo et al., 2010) and Athens, Greece (Kazadzis et al., 2016), especially in winter.

di Sarra et al. (2002), Panicker et al. (2009), and Antón et al. (2011), among others, have shown that an increase of AOD induces a reduction of the UV index (UVI), an effective parameter to quantify the potentially harmful effects of UV radiation. These studies suggested that a unit increase in aerosol optical depth at about 400 nm may produce a significant decrease of UVI which depends on the solar zenith angle and aerosol properties, and may exceed 50%.

This work is aimed at determining for the first time the effect of aerosol optical properties retrieved in Rome on UV radiation, evaluating the role of SSA, AOD and Ångström exponent. The dataset covers the period from March to September of 6 years, from 2010 to 2016. Only Spring and Summer periods were selected, when solar zenith angles smaller than 40° and then higher values of UVI can be analyzed. For SZA>40, as in winter time, the UV index is low, and shows a little range of variability during the day. Therefore the estimation error of UV index, that is about 4-5%, (Schmalwieser et al., 2017) could affect the identification of its variation due to possible aerosol effect. , Aerosol optical properties were



provided by a PREDE-POM sun-sky radiometer of the ESR/SKYNET (www.euroskyrad.net) network,
and the UV index values were measured by a Brewer spectrophotometer.

**2. The site and Instruments**

Rome is a large urban site, with about 3 million inhabitants, located 25 km east of the Tyrrhenian Sea,
in the middle of an undulating plain. The atmosphere is affected by urban emissions as well as by semi-
rural particulates and, especially during the summer season, by sea breeze and long-range desert dust
advection from the Saharan region (e.g., Ciardini et al., 2012).
Long term measurements of aerosol physical and optical properties, columnar ozone content and UV
irradiance (290 -325 nm) are carried out in Rome, on the roof of the Physics Department of Sapienza
University (41.9°N, 12.5°E; altitude 60 m) at the Laboratory of Geophysics. This site is located in the
central sector of the city.
Aerosol properties are retrieved by the observations taken in clear sky conditions by the sun-sky
radiometer PREDE/POM model 01, (hereafter called POM), a narrow band filter photometer able to
perform measurements of direct solar and diffuse sky irradiances at selected wavelengths (315, 400, 500,
675, 870, 940 and 1020 nm) and at 24 scattering angles, in the range [0 –180°] in the almucantar
geometry. The 315 and 940 nm channels are used to retrieve ozone and water vapour columnar content,
whereas the other ones provide information on aerosols. The time resolution is 1 minute for direct
irradiance and 10 minutes for diffuse irradiances. This instrument is part of the European Skynet
Radiometer network (ESR, Campanelli et al., 2012; www.euroskyrad.net) that is a regional subnetwork
of SKYNET (Takamura et al:, 2004); it has been operating in Rome since 2010 up to present. Calibration
is performed monthly by the Improved Langley method (Campanelli et al., 2007), a well-tested "on-
site" procedure that allows to frequently check the instrument status.
UV irradiance and total ozone content have been measured since 1992 at Rome by the Brewer Mk IV
spectrophotometer No.067. This instrument is also operating by the Physics Department of Sapienza
University at the Laboratory of Geophysics in Rome and is part of a European Brewer Network
(EUBREWNET).   The Brewer Mk IV is a single monochromator spectrophotometer specifically
designed to retrieve through a well-defined data processing (Siani et al., 2018) the total column ozone
by measuring solar direct irradiances at selected UV wavelengths in the ozone absorption spectrum (Kerr
et al., 1981). The accuracy of direct-sun measurements of total ozone taken with a well-maintained
Brewer spectrophotometer is 1% (Vanicek, 2006). The  performance of the Brewer instrument for UV
measurements was  controlled every two years till 2014 through intercomparisons to the traveling
reference QASUME UV spectroradiometer operated by Physykalish Meteorologisches Observatorium





Davos/ World Radiation Centre. The mean ratio of Brewer integrated solar UV irradiances to QASUME
is within +3% (see https://www.pmodwrc.ch/en/world-radiation-center-2/wcc-uv/). After that the UV
calibration has been  carried out by IOS using  1000w lamps which are traceable to the QASUME
reference spectroradiometer (Siani, et al:, 2013). The Brewer also measures global spectral irradiances
from 290 nm to 325 nm with a spectral resolution of about 0.5 nm at 0.5 nm steps. UV spectral scans
are performed at Rome every 30 min throughout the day. The Brewer algorithm for the spectral interval
325 -400 nm  assigns a higher weight to the measurement at 324 nm wavelength to compensate for the
missing contribution of wavelengths longer than 325 nm. It has been found that this interpolation method
introduces an error typically <2% in the UV index value for solar zenith angles <70°. (Fioletov et al.,

111  2004).

To complete the characterization of aerosol properties at Rome during summer, results from an intensive
field campaign (URBan Sustainability Related to Observed and Monitored Aerosol – URBS ROMA,
Campanelli et al., 2012 )  conducted in the period June – July 2011 in the same location and aimed to
determine the aerosol direct radiative effect at the surface, were used.  Particulate matter 10 micrometers
or less in diameter (PM10) mass concentrations were collected by using a dual channel sampler
(HYDRA Dual Sampler, FAI Instruments, Fonte Nuova, Rome, IT) equipped with Teflon membrane
filters and quartz fiber filters on the two channels, and PM10 mass concentration was measured on
Teflon filters by gravimetry using an automated microbalance.
The elastic Lidar of the Sapienza University was also operative simultaneously with the other
instruments and, in this study, it was used to discriminate days affected by desert dust.
Finally, during the period under analysis, the following ancillary meteorological parameters have also
been used: atmospheric pressure provided by the Agenzia Regionale per la Protezione Ambientale
(ARPA – Lazio), and cumulated precipitation measured at the station Roma Macao of the Ufficio
Idrografico e Mareografico of Rome, less than 1 km far from the Department of Physics of Sapienza
University.

**3. Methodology**

The POM normalized radiance (that is the radio between the solar diffuse radiance and direct solar
irradiance) is inverted using the Skyrad4.2 pack (Nakajima et al., 1996), which is an official computer
code of the SKYNET network. Signals from the channels centered at the wavelengths of 400, 500, 675,
870, and 1020 nm are analyzed in order to determine AOD, SSA, and Ångström exponent (Ang), the
latter obtained by using all the wavelengths.  In addition, the Ångström exponent is also calculated from
the AOD at 400 and 500 nm (Ang$_{400-500}$) to infer the AOD wavelength dependence in the spectral range



closest to the UV region. Cloud screening and quality check of the retrieved inversions are also
performed. The cloud screening is based on the direct solar irradiance variability in 3 minute time
interval, as explained in Estelles et al. (2012); the quality check of SSA and AOD at 400 nm ($SSA_{400}$,
and $AOD_{400}$, respectively), that is the POM shortest wavelength used in this analysis, is based on the
results from the most recent literature on Skyrad pack. Hashimoto et al. (2012) performed numerical
tests on the $SSA_{400}$ retrievals using the Rstar-6b radiative transfer code (Nakajima and Tanaka, 1986)
and Skyrad pack (versions 4.2 and 5.0) inversions. They obtained $SSA_{400}$ values of about 0.70 for dust-
like and water insoluble aerosol models. A cirrus contamination case, obtained by enhancing the coarse
mode for simulating the presence of ice  particle  types, according to  the  cirrus particles model of the
World Climate Programme report (Deepak and  Gerber, 1983),  provided values varying between 0.71-
0.75. Therefore, in this study measurements of $SSA_{400}$ lower than 0.70 were rejected even if values
between 0.70 and 0.75 could contain information on dust presence during possible cloud contaminated
cases. Hashimoto et al. (2012) also demonstrated that the SSA retrieval by Skyrad4.2 pack is
problematic, since sometimes SSA tends to be unnaturally close to unity, irrespectively of the AOD.
Therefore, inversions where $SSA_{400}$ assumed values $\geq 0.99$ were also rejected. In this work we used only
SSA at 400 nm as absorption estimation parameter because the comparison against retrievals from other
versions of the Skyrad code showed good agreement at this wavelength and discrepancies at the others.
The UVI was introduced in Canada in 1992 (Fioletov, 2010) to represent the potentially harmful effects
of UV radiation in a simple form. UVI is a unit-less quantity determined by multiplying the erythemally
weighted UV irradiances (in $W\ m^{-2}$) over the range 280-400 nm by 40 $m^2 W^{-1}$ (Cost -713, 2000). UVI
values are grouped into exposure category expressing the risk for unprotected skin to Sun exposure.
Typically at mid-latitudes, UVI values at noon vary from 0 to 10, but highest UVI values were
experienced at high altitude (e.g., Casale et al., 2015) and lower latitude sites. Spectral UV irradiances
measured by the Brewer spectrophotometer in clear sky conditions (no clouds over the sun) were used
to retrieve UV index values. The spectral irradiances were processed using the SHICrivm software
(version 3_075) to obtain the biologically effective UV irradiance by weighting the solar irradiances
with a function (action spectrum) representing the effectiveness of UV radiation to produce the
erythemal response in the skin (C.I.E., 1998). The SHICrivm software was also applied to check for
any spectral wavelength shift and spectral anomalies (Slaper et al, 1995) in the UV data. In addition,
since the Brewer MKIV spectrophotometer measures spectral irradiances up to 325 nm, the non-
measured part of the UVA spectrum needed for the calculation of UVI was also extrapolated by the
same software.
Total ozone values ($O_3$) from direct-sun measurements were generated by using Brewer Processing
Software, applying the rejection criteria on ozone values less than 100 DU and greater than 500 DU



(Siani et al., 2018). Yet, individual total ozone values were discarded when standard deviation is above
2.5 DU and ozone air mass (defined as the ratio of the actual ozone path length taken by the direct solar
beam to the analogous vertical ozone path when the Sun is overhead from the surface to the top of the
atmosphere) is above 3.5.

To discern the dependence of UVI only on aerosol characteristics, the UVI dependence on the solar
zenith angle ($\theta$), ozone content, and orbital parameters (varying Earth-Sun distance) must be taken into
account. Therefore, firstly the UVI was corrected for the variation of the Earth-Sun distance and values
were reduced to the mean Sun-Earth distance (Madronich, 1993).  Secondly, only data at two values of
$\theta$, 30° and 40°, were selected.  This criterion excludes winter data, when the solar zenith angle is always
higher than 40° in Rome.  Thirdly, the UVI dependence on total $O_3$ has been removed.  This correction
has been implemented using the Radiation Amplification Factor (RAF) and scaling the UVI to the total
ozone value measured during the day with the lowest $AOD_{400}$ recorded in the entire dataset (303 DU on
September 2, 2014). Infact the effect of ozone on the erythemal UV irradiance may be described as
suggested by Madronich (1993) and Booth and Madronich (1994):
$$\frac{E^*}{E} = \left(\frac{O_3}{O_3^*}\right)^{RAF}, \tag{1}$$

where E and E* are two UV irradiances observations, and $O_3$ and $O_3^*$ their corresponding total ozone
amounts.
Similarly, it is possible to apply the above relationship to UVI:
$$UVI^* = UVI \left(\frac{<O_3>}{O_3^*}\right)^{RAF}, \tag{2}$$

where $<O_3>$ is the diurnal ozone average value, $O_3^*$ is the diurnal ozone average value during the day
with the minimum average $AOD_{400}$, and RAF is assumed to be equal to 1.25, according to di Sarra et al.
(2002). To point out the possible effect of aerosol optical characteristics measured at 400 nm on UVI*,
$AOD_{400}$, $SSA_{400}$, Ang and $Ang_{400-500}$ were analyzed as function of UVI* at the two fixed solar zenith
angles, taking estimations of aerosol parameters and UVI* within ±5 minutes.

Chemical characterization of the collected PM10 dust, during the URBS campaign, was carried out
according to the method reported in Perrino et al. (2009). Briefly, elements were determined on Teflon
filters by X-ray fluorescence (XRF); then the filters were water-extracted and analyzed for their ionic
content by ion chromatography (IC); elemental and organic carbon (EC and OC) were detected on quartz
filters by thermo-optical analysis (NIOSH-QUARTZ temperature protocol). This overall analytical
procedure allows the determination of each individual component typically accounting for more than



1% of the mass amount of PM10 (macro-components: Si, Al, Fe, Na, K, Mg, Ca, chloride, nitrate,
sulfate, ammonium, elemental carbon, organic carbon) and to obtain the mass closure.
PM10 macro-components can be grouped into five clusters to estimate the contribution of the main
macro-sources: SOIL, SEA, SECONDARY INORGANICS, ORGANICS, and TRAFFIC. Details about
the algorithms are reported in Perrino et al. (2014). Briefly, the contribution of SOIL was calculated by
adding the concentration of elements (as metal oxides) generally associated with mineral dust: Al, Si,
Fe, the insoluble fractions of K, Mg, and Ca (calculated as the difference between XRF and IC
determinations), calcium and magnesium carbonate (calculated as the sum of soluble calcium multiplied
by 1.5 and soluble magnesium multiplied by 2.5); SEA was estimated from the sum of $Na^+$ and $Cl^-$,
multiplied by 1.176 in order to take into account minor sea-water components; SECONDARY
INORGANICS were calculated as the sum of non-sea-salt sulphate, nitrate, and ammonium; the
contribution of road TRAFFIC was estimated by adding elemental carbon to an equivalent amount
multiplied by 1.1 in order to consider the contribution of primary organic matter that can be adsorbed on
particles surface; the remaining organic carbon, multiplied by 1.6 to take into account non-C atoms,
constituted the ORGANICS and included both secondary organic species and primary components.
**4. Results**
The analyzed dataset covers the period March – September from 2010 to 2016 (for the last year the series
end in August). Figure 1 shows monthly average $SSA_{400}$, $AOD_{400}$, and Ångström exponent for the period
under examination. Annual means (calculated over the 7 months under study) of $SSA_{400}$ vary between
a minimum value of 0.84±0.08 (observed in 2016) and a maximum of 0.97±0.03 (observed in 2015); the
large SSA decrease in 2016 is also observed by AERONET estimates of SSA at 440 nm (shown with
red points in Figure 1), obtained from measurements taken in TorVergata, a semirural area 14 km south
east of the town. The AERONET inversion is performed according to Dubovik and King (2000) and it
is able to retrieve aerosol optical properties from Sun and sky radiance measurements. In this study we
used level 1.5 data and Version 3 inversion algorithm (Giles et al., 2019). Although the two sites are
slightly different in terms of atmospheric particles optical properties, and the wavelength used for AOD,
SSA and Ångström differs, the agreement between the AERONET and SKYNET properties for the 3
common months in 2016 is significant and the decreasing trend in SSA is visible from AERONET
inversion. The decrease is even stronger from March to May. However, we are not able to identify the
reason for this enhanced aerosol absorption. $AOD_{400}$ mean values range between a minimum of
0.14±0.06 (in 2014) and a maximum of 0.36±0.10 (in 2015; values higher than 0.3 are measured only in
this year for the period under study). The Ångström exponent varies between 0.56±0.29 (in 2012) and
1.49±0.21 (in 2011). The total ozone content values and UVI at local noon are also plotted in Figure.1.



The seasonal ozone behavior is typical of mid-latitude sites, with highest values measured in 2010 and
2016. As expected, UVI has a bell-shape behavior generally peaked in July. The monthly cumulated
precipitation and the monthly average atmospheric pressure are also plotted in the same figure.
Scatter plots of monthly average $AOD_{400}$, $SSA_{400}$, Ang, and UVI versus monthly precipitation (Figure.
2) were performed in order to check if precipitation can affect on average the optical parameters.
The only two parameters showing a slight correlation are $SSA_{400}$ (R=0.30) and UVI (R= -0.60),
highlighting that higher precipitation is associated with higher values of SSA (therefore less absorbent
particulate) and with lower UVI values. These correlations among monthly mean values may be
incidental, or due to the combination of different processes. In particular, we may expect that a higher
occurrence of scattered clouds conditions, corresponding to lower UVI values, may be associated with
periods with high precipitation during short-lived weather spring-summer disturbances. Possible effects
on $SSA_{400}$ may be linked to the possible influence of high humidity conditions, leading to a larger water
content in soluble particles. This is however speculative, and a detailed analysis goes beyond the scope
of this paper.
During June-July 2011 the chemical analysis of the collected PM10 (Figure 3) measured an average
contribution over the entire mass of about 29% of SOIL, 6% of SEA, 23% of SECONDARY
INORGANIC, 28% of ORGANICS and 9% of TRAFFIC components. During the URBS- ROMA
campaign, the elastic Lidar showed the presence of significant events of desert dust transport, the
strongest observed during the days highlighted in orange in Figure 3. It must be considered that in the
days flagged as "dusty", dust can remain at a higher level and not measurable at ground (this is the case
of 3 and 18 July). Conversely, sometimes a lot of aerosol is visible at ground level but it was not possible
discriminating the presence of desert dust from the local SOIL component (this is the case of July 2 and
17).. The atmosphere over Rome, during summer, generally is characterized by a contribution of SEA
comparable with the TRAFFIC, or even greater during days with no desert dust advection. The
absorption capability of these two components is very different: in the Rstar radiative transfer model
(Nakajima and Tanaka 1986), at 413 nm the imaginary part of marine aerosol refractive index is
$2.42x*10^{-8}$, whereas for soot, that is the fundamental material characterizing the TRAFFIC component,
is $4.57x10^{-1}$. The mineral component has a refractive index of $7.95x10^{-3}$ at the same wavelength. It is
therefore expected that modulation of the concentration of the three co-existent materials, can strongly
affect the absorption capability of the atmosphere over Rome.



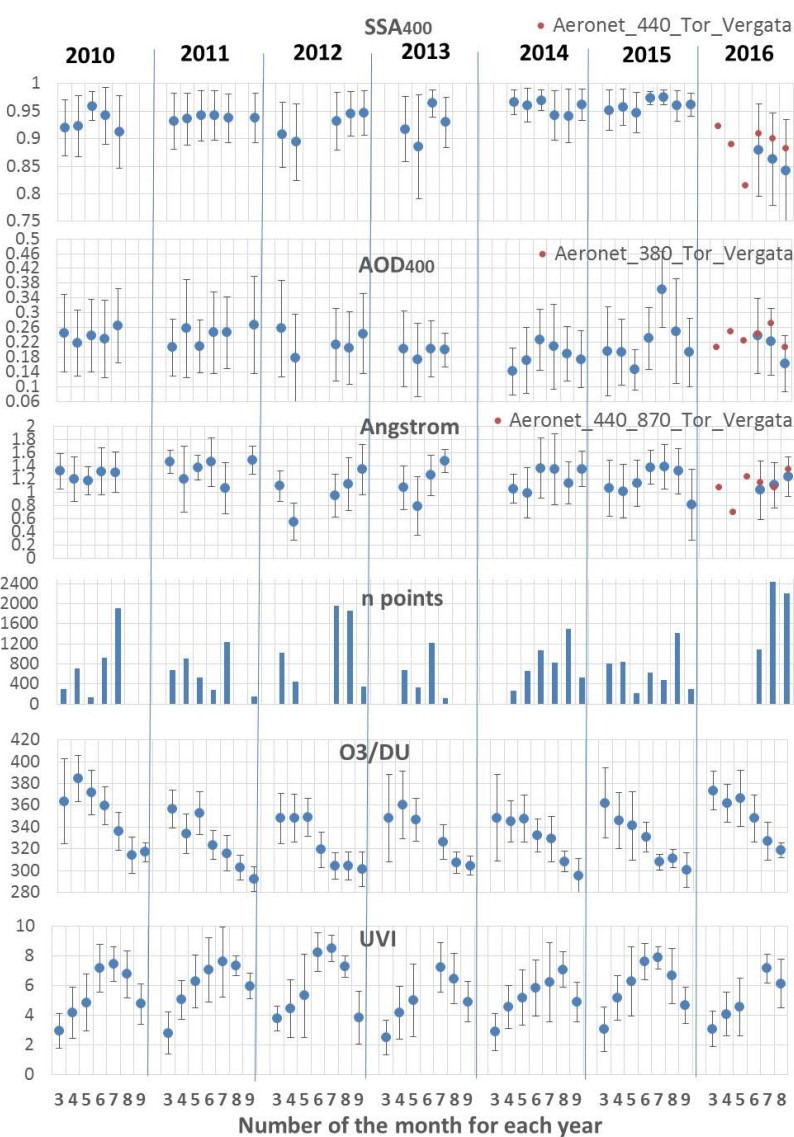



Figure 1: Monthly averages of $SSA_{400}$, $AOD_{400}$, Ångström exponent, cumulated precipitation, total $O_3$,
UVI, and atmospheric pressure for each year from 2010 to 2106. The number of points refers to the data
used to retrieve the aerosol parameters. Error bars are the standard deviation.





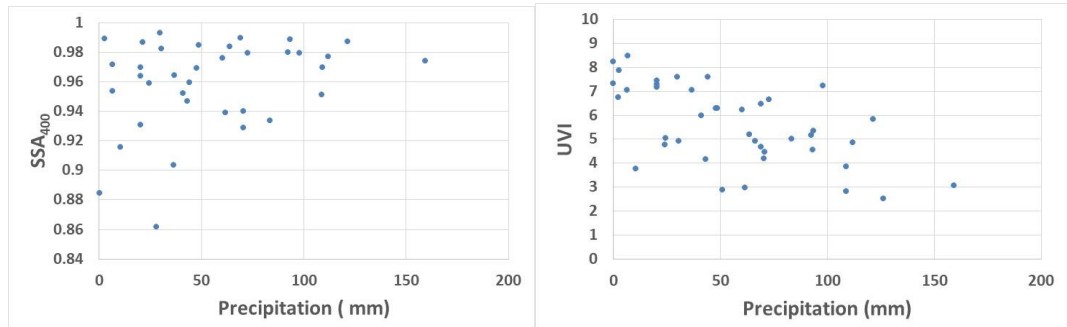


Figure 2: monthly average of SSA$_{400}$ (left) and UVI (right) versus monthly precipitation



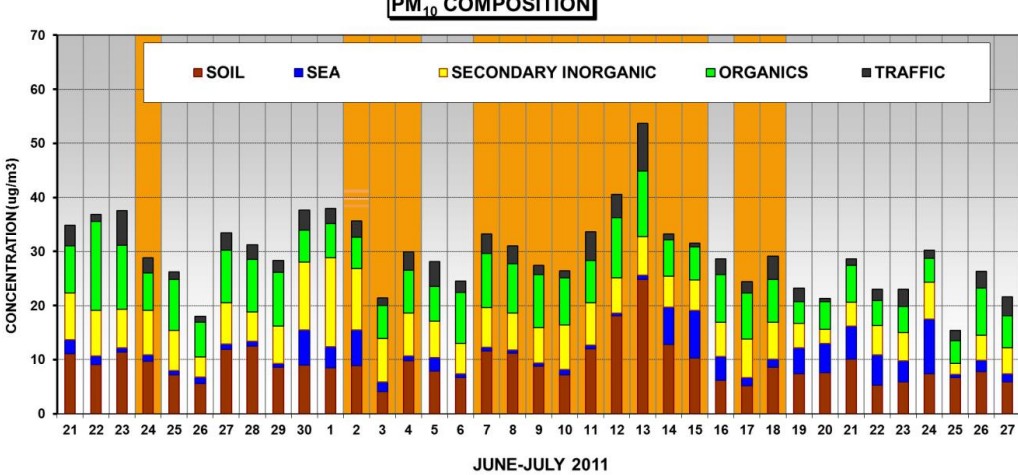


Figure 3: Concentration of the components of PM10 collected in Rome from 21 June to 27 July 2011 as
derived from chemical analyses. Orange columns represent days affected by the passage of desert dust,
as measured by Lidar.

A statistical analysis of daily means of SSA$_{400}$, AOD$_{400}$ and Ångström exponent with the percentage
contribution of each chemical component, has been performed in order to connect optical properties and
chemical analysis. In fact, assuming that the in situ measurements are representative of the entire column,
their variation affects particles refractive index and particles dimensions, and consequently their
absorption capability and Ångström exponent. Scatter plot of SSA$_{400}$ versus the SOIL component
(Figure 4) shows a slight negative correlation (R= -0.54), whereas no other correlation is visible for the
other components and other optical and physical parameters. This result underlines that in situ





measurements may not provide information correlated with the columnar properties, because optical and
physical properties at the ground may differ from those of the entire column. Therefore, both information
must be used complementarily for understanding the radiative effects of such a mixture of different
components.

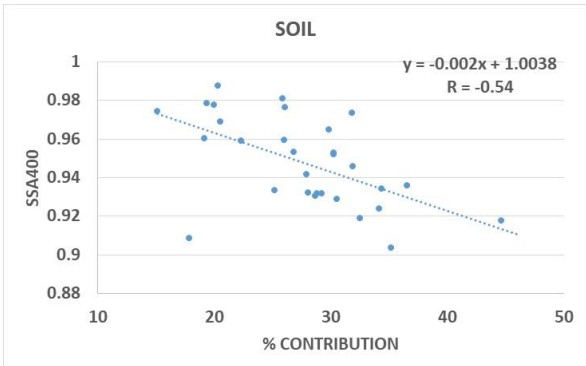


Figure 4. Behaviour of $SSA_{400}$ versus the percentage contribution of SOIL component as retrieved during
the URBS campaign.

Assuming that relations between aerosol composition and their optical properties, measured during
summer 2011, are comparable in the last years, they can be considered as representative of the summer
period 2010-2016 studied in this paper.
In order to point out the possible effect of aerosol optical characteristics measured at 400 nm on $UVI^*$,
the $AOD_{400}$, $SSA_{400}$, Ang, and $Ang_{400-500}$, were analyzed as function of $UVI^*$, at the two selected values
of solar zenith angle. Figure 5 shows the frequency distributions of the number of measurements for
each of the two angles. $\theta=30°$ is more representative of the warmest months, whereas $40°$ covers a wider
period.

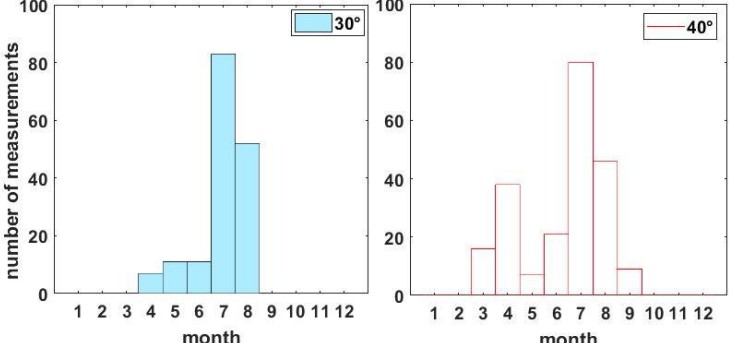


Figure 5: Number of measurements available for each zenith angle.






The dependency of $UVI^*$ on $AOD_{400}$, $SSA_{400}$, Ang and $Ang_{400\_500}$ for 30° and 40° solar zenith angles are
shown in Figure 6, colored for different values of $SSA_{400}$ or $AOD_{400}$. A clear linear decreasing trend of
$UVI^*$ when increasing $AOD_{400}$ is evident. The slope in these graphs corresponds to the UVI* radiative
forcing efficiency, i.e., the change in UVI* produced by a unit change in AOD. The slope is more
pronounced at the smaller solar zenith angle (Table I), as already found by previous studies (di Sarra et
al., 2008; Antón et al., 2011). No clear dependence of $UVI^*$ on $SSA_{400}$ or on Ångström exponents can
be noticed.  If existent, it is expected to be masked by the dependency on AOD, which is the primary
parameter affecting the surface irradiance.
To investigate in more detail, the entire dataset was divided in three groups of $Ang_{400\_500}$, below 0.8,
between 0.8 and 1.7, and above 1.7, and in two groups of $SSA_{400}$, smaller and larger than 0.85,
respectively. The values separating the different groups were determined according to the frequency
distributions of the two variables for the entire investigation period, shown in Figure 7. Scatter plots and
linear fits of UVI* versus the two variables, for each group, were performed and points with a distance
greater than 2σ from the regression line (nout), with σ the standard deviation of the residuals, were
rejected.
The dependence of $UVI^*$ on AOD for the three classes of $Ang_{400-500}$ is shown in Figure 8, colored for
different values of $SSA_{400}$, and in Table I.  The slope is generally larger for smaller values of $Ang_{400-500}$,
similarly to what found by Antón et al. (2011).  At 30° the other two classes of $Ang_{400-500}$ have a very
similar slope, differing of 0.15 that is below its uncertainty estimation from the fit. Conversely at 40° an
intermediate value of the slope is found for $Ang_{400-500} \geq 1.7$; this value appears essentially driven, for
both the zenith angles, by cases with low SSA and low AOD, which might be attributed to a possible
influence from combustion particles characterized by small size and high absorption (see, e.g., Pace et
al., 2005).  A similar dependency on the Ångström exponent was found by di Sarra et al. (2008) when
considering the forcing efficiency over the whole shortwave spectral range. The smallest slope is
associated to the $0.8<Ang_{400-500}\leq1.7$, range which is characterized by a larger mixture of absorption
capabilities.
The Ångström exponent in Rome varies between about 0.5 and 1.8 (Figure 7), with a typical range of
variability of 1.3.  The estimated effect of the Ang variability can be determined by considering the slope
difference among the different values of Ang, which is of the order of 1.5 at 30° solar zenith angle.  The
corresponding change of UVI* is about 2.
Figure 9 shows the scatter plots of $UVI^*$ vs $AOD_{400}$ for $SSA_{400} <0.85$ (left side) and $SSA_{400} \geq 0.85$ (right
side), with a colour scale for different values of the Ångström exponent at the two zenith angles.. For
solar zenith angles 30° (Table II) the slope of $UVI^*$ versus $AOD_{400}$ is larger for $SSA_{400}\geq0.85$, increasing





of about 67% going from -1.77 to -2.96. This increase is significant, since it is greater than the
uncertainty of the estimated slope. For solar zenith angles 40° the increase is about 9%, going from -
1.42 to -1.55, but in this case it is comparable with the estimated uncertainties of the slope, varying from
15% for $SSA_{400} < 0.85$, to 7% for $SSA_{400} \geq 0.85$. This result is opposite to what Antòn et al. (2011) found
in Granada, Spain, where, as expected, stronger aerosol absorption leads to a large surface forcing
efficiency.
Looking at the UVI* versus $AOD_{400}$ or the UVI* versus $SSA_{400}$ scatter plots in Figure 6 it is evident that
for both solar zenith angles (but mostly at the smaller one) less absorbing particles (higher $SSA_{400}$)
correspond to higher $AOD_{400}$. This is also confirmed by the mean and median $AOD_{400}$ values calculated
over all the years in the months analyzed in Rome (Table III) with the additional information that higher
$AOD_{400}$ are also characterized by greater particles ($Ang_{400-500} < 0.8$). This is probably due to the presence
of SOIL and SEA salt in the atmosphere, as highlighted during URBS.
As shown in Figure 7, SSA varies between about 0.75 and 1.0, for a variability range of 0.25. The slope
difference among the different values of SSA about 1, and a rough estimate of the corresponding change
of UVI* is of about 0.25. This value is much smaller than the expected effect produced by Ang that is
a change of about 2. Thus, it is very likely that the effect of variations of single scattering albedo may
be masked by concomitant changes of Ang.

| θ=**30°** | Slope (m) | Intercept (q) | R | θ=**40°** | Slope (m) | Intercept (q) | R |
|---|---|---|---|---|---|---|---|
| $Ang_{400-500} < 0.8$ | -3.73±0.31 | 8.04 | -0.96 | $Ang_{400-500} < 0.8$ | -2.46±0.34 | 6.00 | -0.87 |
| $0.8 \leq Ang_{400-500} < 1.7$ | -2.28±0.24 | 7.82 | -0.77 | $0.8 \leq Ang_{400-500} < 1.7$ | -1.38±0.11 | 5.68 | -0.78 |
| $Ang_{400-500} \geq 1.7$ | -2.13±0.37 | 7.76 | -0.78 | $Ang_{400-500} \geq 1.7$ | -1.62±0.24 | 5.62 | -0.83 |


Table I: The slope, intercept and correlation coefficient for the linear fit of UVI* vs $AOD_{400}$, in three
cases: data selected for $Ang_{400-500} < 0.8$; $0.8 \leq Ang_{400-500} < 1.7$; $Ang_{400-500} \geq 1.7$, for the two zenith angles

| θ=**30°** | Slope (m) | Intercept (q) | R | θ=**40°** | Slope (m) | Intercept (q) | R |
|---|---|---|---|---|---|---|---|
| All data | -1.97±0.21 | 7.80 | -0.65 | All data | -1.36±0.14 | 5.68 | -0.60 |
| $SSA_{400} < 0.85$ | -1.77±0.21 | 7.71 | -0.77 | $SSA_{400} < 0.85$ | -1.42±0.22 | 5.61 | -0.73 |
| $SSA_{400} \geq 0.85$ | -2.96±0.21 | 8.17 | -0.89 | $SSA_{400} > 0.85$ | -1.55±0.11 | 5.76 | -0.82 |


Table II: slope, intercept and correlation coefficient for the linear fit of UVI* vs $AOD_{400}$, in three cases:
all the dataset, data selected for $SSA_{400} < 0.85$ and $SSA_{400} \geq 0.85$ for the two zenith angles.



|  | AOD$_{400}$ at θ=**30°** | | AOD$_{400}$ at θ=**40°** | |
|---|---|---|---|---|
|  | Mean ± std | median | Mean ± std | median |
| SSA$_{400}$<0.85 | 0.186±0.099 | 0.185 | 0.200±0.095 | 0.187 |
| SSA$_{400}$≥0.85 | 0.296±0.118 | 0.274 | 0.262±0.135 | 0.249 |
| Ang400_500<0.8 | 0.345±0.134 | 0.330 | 0.218±0.129 | 0.174 |
| Ang400_500>=1.7 | 0.117±0.066 | 0.105 | 0.155±0.088 | 0.124 |


Table III: mean and median AOD$_{400}$ values calculated over all the years in the months analyzed in Rome,
separately for different classes of SSA and Ang.


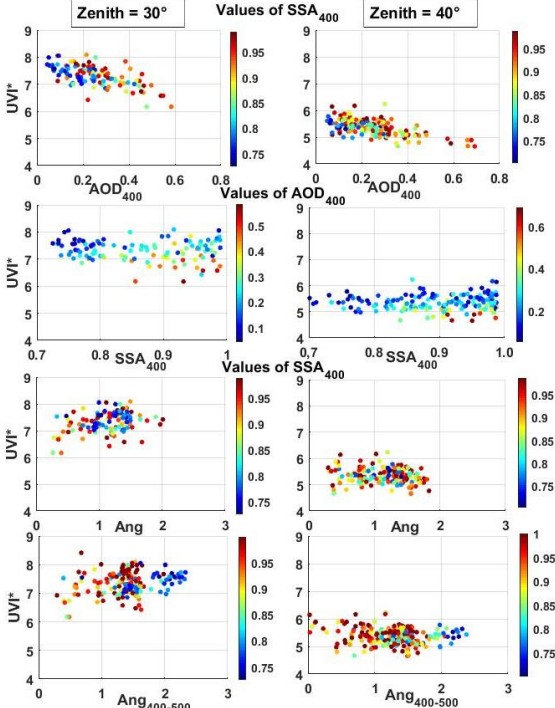


Figure 6.  Scatter plot of UVI$^{*}$ vs AOD$_{400}$ (top), SSA$_{400}$ (middle), and Ang and Ang$_{400-500}$(bottom) for
the solar zenith angles of 30° (left) and of 40° (right). The colors represent the values of SSA$_{400}$ (first,
third and fourth rows) and AOD$_{400}$ (second row).

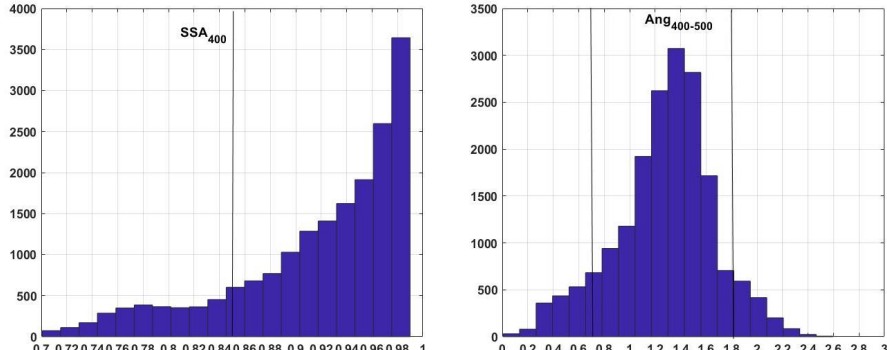


Figure 7. Frequency distributions of $SSA_{400}$ (left) and $Ang_{400-500}$ (right) for the entire investigation


period. The threshold values separating the different classes are highlighted with vertical black lines.



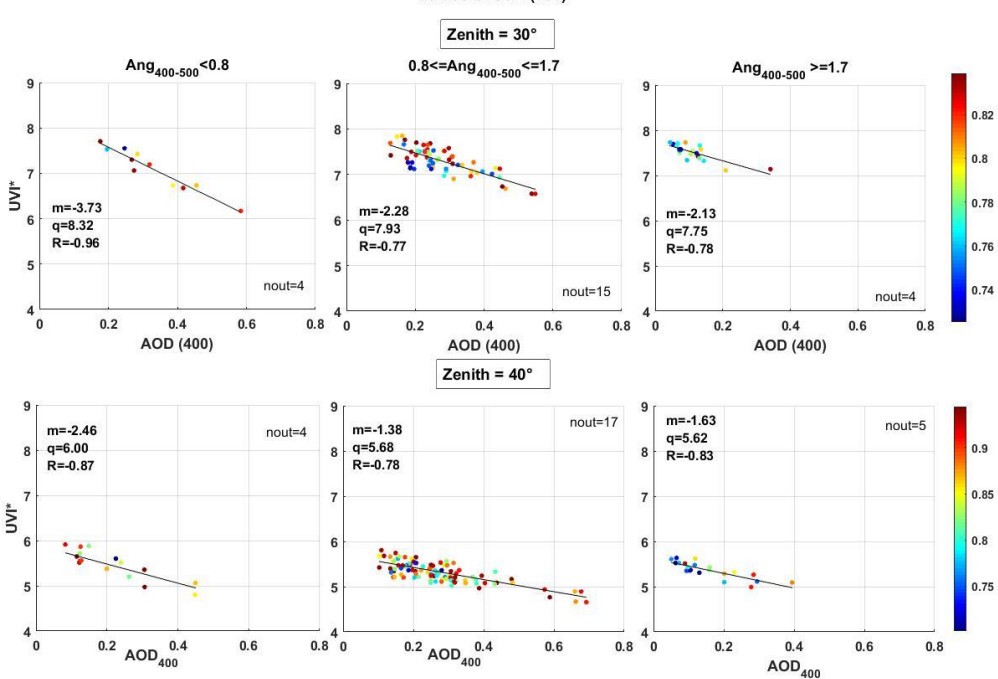


Figure 8: scatter plot of $UVI^*$ vs $AOD_{400}$ for three groups of $Ang_{400-500}$ (left, middle, right) and two


solar zenith angles (top, bottom). The colors represent the values of $SSA_{400}$. nout is the number of


rejected outliers.




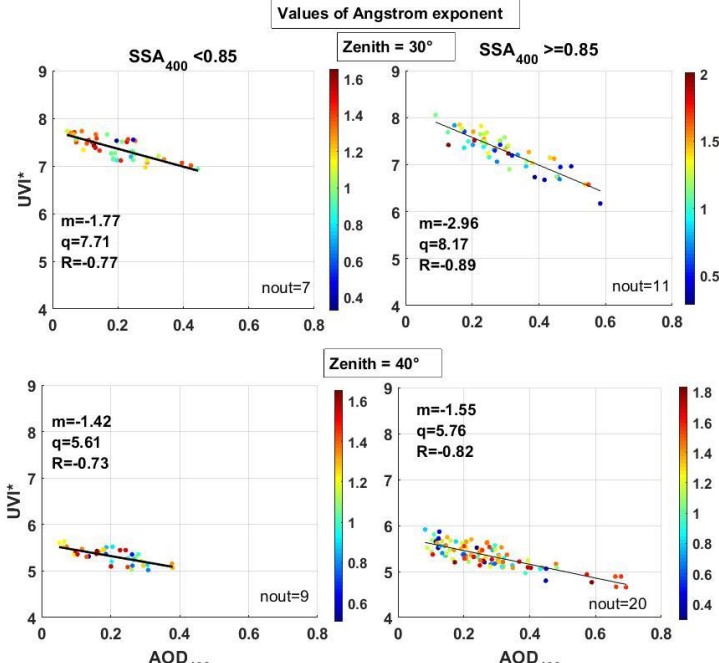

Figure 9: scatter plot of UVI* vs AOD400 for two groups of SSA400 (right and left) and two solar
zenith angles (top and bottom). The colour scale refers to the values of Ang. nout is the number of
rejected outliers.

**5. Conclusions**

The aerosol optical characteristics in the urban area of Rome were retrieved for a period of 7 years, in
the months from March to September 2010-2016. The impact of SSA, AOD at 400 nm, and Ångström
exponent on the UV index has been analyzed. The evolution of UVI*, which is the measured UV index
corrected for total ozone changes and scaled at the mean Sun-Earth distance, was studied d with respect
to $AOD_{400}$, $SSA_{400}$, and Ångström exponent calculated using all the wavelengths (Ang) and only AOD
at 400 and 500 nm. Data at two fixed values of the solar zenith angle were selected in order to point out
the possible effect of aerosol optical characteristics measured at 400 nm on UVI*.

A clear linear decreasing trend of UVI* when increasing $AOD_{400}$ was found, with a more pronounced
slope at the smaller solar zenith angle, as already shown by previous studies. The dependence of UVI*
on $SSA_{400}$ and Ångström exponents is probably masked by the dependency on AOD, which is the
primary parameter affecting the surface irradiance. The entire dataset was also analyzed separately for
different absorption properties (by fixing a threshold value for $SSA_{400}$) and for different aerosols
dimensions (by fixing threshold values for $Ang_{400-500}$). The surface forcing efficiency, provided by the



decreasing trend of UVI* with $AOD_{400}$, was found greater for larger particles. In Rome these particles,
having small Ångström exponent values, are generally less absorbing since related to the presence of
SOIL and SEA components in the atmosphere. Moreover the former contribution is much higher in
summer months (as highlighted from the chemical characterization of suspended particulate matter over
Rome during the URBS ROMA intensive field campaign held in 2011) because of the numerous
episodes of Saharan dust transport. The result is that the effect of the Angstrom exponent on the incoming
UV radiation could mask the dependence on the SSA.
The general behavior observed for the five macro-sources (SOIL, SEA, SECONDARY INORGANIC,
ORGANICS and TRAFFIC) during summer 2011 has been assumed not substantially changed in the
last years, and  the variations in the absorption capability of the atmosphere over Rome were attributed
to the different absorption characteristics of the macro- components and their modulation of
concentration in the atmospheric mixture.
A better understanding of the impact of aerosol optical properties in Rome on UVI* can be done in the
next future using measurements of direct and diffuses solar radiation at 340 nm, instead of 400, available
at the ESR Rome site from 2018. Also the use of different versions of the Skyrad code (as version 5.0)
can improve the retrieval of the SSA wavelength dependence, making possible the calculation of the
Absorption Ångström Exponent for a better characterization of the absorption properties.

**6. Acknowledgements**:
We thank Gian Paolo Gobbi and collaborators for establishing and maintaining the Rome–Tor Vergata
AERONET site used in this investigation. We also thanks ARPA-LAZIO for providing meteorological
data over Rome.

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
