# Peer review of "Aerosol optical characteristics in the urban area of Rome, Italy, and their impact"

_Atmospheric Measurement Techniques, 2019_

## Referee Comment (RC1) · Anonymous Referee #2 · 4 Oct 2019

A) General comments The manuscript is a description of the analysis of atmospheric Aerosols and solar UV measurement in Rome, Italy. The targets are scientists interested in the both, the relation between aerosols and UV radiation and the measurement of the aerosols in the city of Rome itself. Next to the detailed explanation of the measurements, the data of the years 2010-2016 have been analyzed. Altogether this results in a high-quality analysis and a nice study of the relationship of aerosols and UV radiation in a city with significant pollution (aerosols) and high level of UV (Italy). The work is well presented and in good quality both in writing and presenting. However, the author tends to very long sentences which makes the reading and understanding more difficult.

B) Specific scientific comments All technical parts of the measurements and analysis are well described. The following additionally point should be discussed to enhance the quality of the manuscript: 1. Line 55: In the Introduction it was explicitly mentioned that "especially in Winter" a good relation between aerosols and UV was found. However, in this study only Spring and Summer month were used. The author should at least discuss why their data with SZA>40 is not usable. 2. Line 97: The Uncertainty of the total ozone measurements is given with 1%. However, no estimate is given for the actual UV measurements. Especially a discussion of the uncertainty of the extrapolation in the UVA range (325nm-400nm) is missing (measurement only up to 325nm!). 3. Line 101: "In addition several tests are performed". Well, the IOS intercomparison is mainly used for the Brewer Ozone measurements. The traceability of the UV measurements is either calibrated using irradiance standards or obtained through intercomparisons to reference spectroradiometers. If any of those action are available for Brewer #067 it should be mentioned and referenced in the paper. 4. Line 218: "AERONET inversion" should be explained. 5. Line 241: "In these days a substantial decrease..." –Figure 3 shows sometimes a small decrease but also an increase of sea and soil (1 to 2 July). In the third event the soil components increased from 12 to 25 %! 6. Figure 1: Errors bars indicate only the measurements uncertainty (?) but not the total expanded uncertainty of the measurements. See also comment 2. 7. Normalization of UVI: RAF is according to the referenced paper (Di Sarra 2002) of high uncertainty (between 0.8 and 1.44. Taking 1.25 should be justified in more detail and added to the (missing) uncertainty budget.

C.) Presentation The manuscript is clearly structured. Minor modifications are recommended to improve the quality of the paper: Line 166: Subscript 0 of Theta_0 is irritating. Line 194: "PM" -> probably "PM10" is correct at this position. Line 226 – 228: Good example of a very confusing long sentence. Figure 5: The two different bar-plots for SZA=30 deg and SZA=40 deg cannot be distinguished. Line 409: "direct solar radiation" or "direct and diffuse solar radiation"? Figure 1 is overloaded. "n points" should be part of the uncertainty budget, "precipitation and pressure" is not used and these

graphs don't add relevant information. Units in the figure axis labels should be labeled as "/unit" to have a dimensionless number and not "(unit)". Figure is using a different labelling (% CONTRIBUTION").

Typos: line 336: "whit" -> "with" line 348: (Table II): Theta=40 deg -> 40 in bold

---

## Referee Comment (RC2) · Anonymous Referee #3 · 30 Oct 2019

General comments:

The manuscript describes the aerosol optical characteristics in the urban area of Rome for the time period 2010-2016. The impact of aerosol single scattering albedo, aerosol optical depth and Ångström component on the UV index are analyzed. Chemical characterization of urban PM10 samples from a field campaign was performed, and the contribution of main macro-sources was evaluated. The data set is analyzed for the first time and this kind of analyses of aerosol optical characteristics in the urban area of Rome is novel.

Specific comments:

[Figure]

Major comments: You write in the abstract that "PM macro-components were grouped in order to evaluate the contribution of the main macro-sources (SOIL, SEA, SECONDARY INORGANIC, ORGANICS and TRAFFIC) and the analysis of the modulation of their concentration was found to strongly affects the absorption capability of the atmosphere over Rome." However, I don't find clearly explained in the Results Chapter of the manuscript the statement "the analysis of the modulation of their concentration was found to strongly affects the absorption capability of the atmosphere over Rome." Please explain more clearly the connection between the PM macro-components and the absorption capability of the atmosphere over Rome. From the Results I mostly understand from page 10, lines 285-286, that "Scatter plot of SSA400versus the SOIL component (Figure4) shows a slight negative correlation (R= -0.54), whereas no other correlation is visible for the other components and other optical and physical parameters. " In case it is related to the theoretical calculations of page 8, lines 260-265, please explain in more details in Methodology how you have used the model.

Add to the methodology the use of Lidar for detection of Sahara dust events.

Minor comments: Abstract Page 1 L28: "...is the primary parameter affecting the surface irradiance. . ..". Please specify that it is for clear sky at Rome. In some other sites, total ozone can the the primary parameter for clear skies. And if not clear skies, then cloudiness has an important role.

Page 1, line 38. Can you specify why the aerosol influence on UV is still uncertain.

Page 4, lines 108-111, Why is this mentioned here, if UVI used in the study is calculated using SHICRIVM? I suggest to remove those line.

Page 5, lines 142-148, for people who are not familiar with cloud screening of aerosol measurements, the explanation is not clear. Please reformulate the reason for rejecting SSAs lower than 0.70.

Page 5, line 158, Which are the highest UVI values (give some numbers).

Page 5, line 159, How the clear sky spectra were selected?

Page 7, lines 224-225. It would be good to show the AERONET data also for the other years. Then it would be easier to the agreement/disagreement between the two instruments.

Page 7, lines 225-226, Please move the AERONET explanation into the Section Methodology. Including the use of different wavelengths than for POM.

Page 7, line 230, How did you defined that agreement is significant?

Page 7, line 235, Monthly averages of the total ozone content values and . . ..

Page 8, line 236, Please explain which kind of seasonal variability (higher in spring. . .) and give the highest values.

Page 8, line 238, I don't see the cumulated precipitation and pressure in the Figure.

Page 8, line 239, only the SSA400 and UVI is plotted in Figure 2. Not the AOD or Ang.

Page 8, line 245, I don't understand the explanation, as aren't the UVI measurements performed under clear skies?

Page 9, Figure 1 caption, Explain the red point also in the Figure caption. Add "monthly means" and for UVI → local noon.

Page 10, Figure 2, local noon UVI under clear sky? Page 12, lines 314 and 315: Why not to try to analyse the aerosol absorption optical depth AAOD (1-SSA)*AOD?

Page 12, lines 316-319: What was the criteria for the chosen values to be used to make the division into subgroups?

Page 12, line 337: How did you calculate the corresponding change of UVI* to be about 2?

Page 13, line 345: Can you give any reasons why your results differ from those of Anton et al. 2011?

Page 13, line 354: There is a missing verb in the sentence + add at 30 degree, if so.

Figures 8 and 9, Move the heading of the color panels on the top of the color panel.

Page 16, line 398, Where did you show results of analysis using AOD at 500 nm?

Page 16, line 401-403, Here again, why not to study the AAOD?

Page 16, lines 413-415. What supports your assumption that the five macro – sources have not changed in the last years? And as I wrote in the General Comment, I don't understand what supports the sentence "the variations in the absorption capability of the atmosphere over Rome were attributed to the different absorption characteristics of the macro-components and their modulation of concentration in the atmospheric mixture."

Technical corrections:

In general the text contains long sentences, which are difficult to follow. Paragraph breaks are missing, e.g. in the section Results.

Page 5, New paragraph cut between lines 152 and 153. Same for line 158 before Spectral UV...

---

## Referee Comment (RC3) · Anonymous Referee #1 · 11 Nov 2019

The authors present their findings on the impact of aerosol optical properties on the UV Index a) Main comment: The title promises results in respect to the UV Index. Therefore it would be valuable to find the most important findings in respect to the UV Index already in the abstract. (PS: it is not really interesting where the Brewer is mounted.) Also the "conclusions" should focus on influence of aerosols on UV Index.

b) Minor comments: l.16-17: 2010-2016 is it 6 or 7 years? l.20: Optical data (abstract and introduction): please provide details l.22: delete l.23: PM10 (abstract and elsewhere): provide definition l.30: SOIL and SEA type aerosols? l.82+l.122: . . . and at several scattering angles in the almucantar geometry: please clarify, what is the

difference/meaning, otherwise delete l.123: . . . official code ... is it computer code? l.147: Cost-713 not found in references l.163: . . . ozone air mass. . ... Please explain. I think that this a technical term is known in the Brewer/Dobson community but rather unknown for other readers. Figures: please check labelling of all figures: e.g. Fig.1 axis (second last panel), labelling of color scales,. . ..

---

## Author Comment (AC1) · 10 Jan 2020

**Reviewer #2**

A) General comments.

The manuscript is a description of the analysis of atmospheric Aerosols and solar UV measurement in Rome, Italy. The targets are scientists interested in the both, the relation between aerosols and UV radiation and the measurement of the aerosols in the city of Rome itself. Next to the detailed explanation of the measurements, the data of the years 2010-2016 have been analyzed. Altogether this results in a high-quality analysis and a nice study of the relationship of aerosols and UV radiation in a city with significant pollution (aerosols) and high level of UV (Italy). The work is well presented and in good quality both in writing and presenting. However, the author tends to very long sentences which makes the reading and understanding more difficult.

B) Specific scientific comments.

All technical parts of the measurements and analysis are well described. The following additionally point should be discussed to enhance the quality of the manuscript:

1. Line 55: In the Introduction it was explicitly mentioned that "especially in Winter" a good relation between aerosols and UV was found. However, in this study only Spring and Summer month were used. The author should at least discuss why their data with SZA>40 is not usable.

For SZA>40, as in winter time, the uncertainty on the irradiances measured by the Brewer increase due to effects as straight light interference (Bais and Zerefos, 1996) and angular response error (Antòn et al., 2008). Therefore an enhancement of the estimated error of UV index, which is about 4-5%, (Schmalwieser et al., 2017) is also expected. This could affect the identification of its variation caused by aerosol effect, because the UV index is low at SZA>40 and shows a little range of variability during the day.  We added this sentence in the introduction.

2. Line 97: The Uncertainty of the total ozone measurements is given with 1%. However, no estimate is given for the actual UV measurements. Especially a discussion of the uncertainty of the extrapolation in the UVA range (325nm-400nm) is missing (measurement only up to 325nm!).

The SHICrivm algorithm, used to obtain the biologically effective UV irradiance, as explained in the section 3, compensates for the missing contribution of wavelengths longer than 325 nm. Based on considerations for similar corrections in the Brewer operating software (Fioletov et al., 2004), we estimate an uncertainty <2% in the UV index value for solar zenith angles <70° due to this extrapolation. This sentence has been added in section 2.

3. Line 101: "In addition several tests are performed". Well, the IOS intercomparison is mainly used for the Brewer Ozone measurements. The traceability of the UV measurements is either calibrated using irradiance standards or obtained through intercomparisons to reference spectroradiometers. If any of those action are available for Brewer #067 it should be mentioned and referenced in the paper.

The performance of the Brewer instrument for UV measurements was controlled every two years until 2014 through intercomparisons to the traveling reference QASUME UV spectroradiometer operated by Physykalish Meteorologisches Observatorium Davos/ World Radiation Centre. The mean ratio of Brewer integrated solar UV irradiances to QASUME is within +3% (see https://www.pmodwrc.ch/en/world-radiation-center-2/wcc-uv/). After that, the UV calibration has been carried out by IOS using 1000W lamps, which are traceable to the QASUME reference spectroradiometer. Siani, A.M., Modesti, S., Casale, G.R., Diemoz, H., Colosimo, A. Biologically effective surface UV climatology at Rome and Aosta, Italy (2013) AIP Conference Proceedings, 1531, 903-906, DOI:  10.1063/1.4804917. Both this sentence and the Reference have been added in section2.

4. Line 218: "AERONET inversion" should be explained.

The following sentence has been added in section 4: The AERONET inversion, performed according to Dubovik and King (2000), is able to retrieve aerosol optical properties from Sun and sky radiance measurements. In this study, we used level 1.5 data and Version 3 inversion algorithm (Giles et al., 2019).

5. Line 241: "In these days a substantial decrease. . ." –Figure 3 shows sometimes a small decrease but also an increase of sea and soil (1 to 2 July). In the third event the soil components increased from 12 to 25 %!

We stated that during dust episodes "a substantial decrease of the contribution of SEA and an increase of SOIL components were observed, whereas the others remain quite stable". This happens if we perform an average over all the days recognized as affected (or not) by dust following Lidar profiles.

It must be considered that in the days flagged as "dusty", dust can remain at a higher level and not measurable at ground (this is the case of 3 and 18 July). Conversely, sometimes a lot of aerosol is visible at ground level but it was not possible discriminating the presence of desert dust from the local SOIL component (this is the case of July 2 and 17).

Therefore is order to avoid misunderstanding in the interpretation, we deleted the sentence, but we added the above considerations in section 4.

6. Figure 1: Errors bars indicate only the measurements uncertainty (?) but not the total expanded uncertainty of the measurements. See also comment 2.

Error bars in Figure 1 are the standard deviation related to the monthly average that are larger than the instrumental uncertainties. We improved in the caption the definition of the error bars

7. Normalization of UVI: RAF is according to the referenced paper (Di Sarra 2002) of high uncertainty (between 0.8 and 1.44. Taking 1.25 should be justified in more detail and added to the (missing) uncertainty budget.

di Sarra et al. 2002 (Figure 8) retrieved values of RAF after correcting for the influence of co-varying aerosol optical depth. They retrieved values between 1.0 and 1.2 at 30° and 40° solar zenith angle when considering all aerosol conditions. As discussed in the paper, these values are affected by different processes (the wavelength dependence of the aerosol sensitivity, the interdependence between ozone and aerosol, possibly through increased ozone absorption following enhanced scattering by aerosols, ozone and aerosol vertical distributions). The values of 1.25 was derived from UVSPEC radiative transfer model calculations where the aerosol amount was kept fixed. This value is also in agreement with various other determinations of the ozone RAF (e.g., De Luisi and Harris, 1983; McKenzie et al., 1991; Kerr and McElroy, 1993). However a sensitivity study of UVI* on RAF variation from 1 to 1.25 has been performed over all the dataset showing an average decreasing of UVI* of about 1.4% that is within the declared uncertainty of 4-5%, (Schmalwieser et al., 2017).

This has been added in the text

C.) Presentation

The manuscript is clearly structured. Minor modifications are recommended to improve the quality of the paper:

Line 166: Subscript 0 of Theta_0 is irritating. Removed the subscript

Line 194: "PM" -> probably "PM10" is correct at this position. Corrected

Line 226 – 228: Good example of a very confusing long sentence. The sentence has been changed in " Scatter plots of monthly average $AOD_{400}$, $SSA_{400}$, Ang, and UVI versus monthly precipitation (Figure. 2) were performed in order to check if precipitation can affect on average the optical parameters".

Figure 5: The two different bar-plots for SZA=30 deg and SZA=40 deg cannot be distinguished. The plots were separated

Line 409: "direct solar radiation"or "direct and diffuse solar radiation"? corrected

Figure 1 is overloaded. "n points" should be part of the uncertainty budget, "precipitation and pressure" is not used and these C2 graphs don't add relevant information. We prefer n points visible inside the plot. We deleted pressure and precipitation

Units in the figure axis labels should be labeled as "/unit" to have a dimensionless number and not "(unit)". We changed O3(DU) in O3/DU, since it is the only quantity having a unit

 Figure is using a different labelling (% CONTRIBUTION"). We think the reviewer refers to Figure 3 where the concentration are plotted and not the contribution. We corrected the sentence in the test from "During June-July 2011 the chemical analysis of the collected PM10 (Figure 3) **showed** an average contribution" to " During June-July 2011 the chemical analysis of the collected PM10 (Figure 3) **measured** an average contribution"

Typos: line 336: "whit" -> "with" corrected

line 348: (Table II): Theta=40 deg -> 40 in bold corrected

---

## Author Comment (AC2) · 10 Jan 2020

General comments:
The manuscript describes the aerosol optical characteristics in the urban area of Rome for the time period 2010-2016. The impact of aerosol single scattering albedo, aerosol optical depth and Ångström component on the UV index are analyzed. Chemical characterization of urban PM10 samples from a field campaign was performed, and the contribution of main macro-sources was evaluated. The data set is analyzed for the first time and this kind of analyses of aerosol optical characteristics in the urban area of Rome is novel.

Specific comments:

Major comments: You write in the abstract that "PM macro-components were grouped in order to evaluate the contribution of the main macro-sources (SOIL, SEA, SECONDARY INORGANIC, ORGANICS and TRAFFIC) and the analysis of the modulation of their concentration was found to strongly affects the absorption capability of the atmosphere over Rome." However, I don't find clearly explained in the Results Chapter of the manuscript the statement "the analysis of the modulation of their concentration was found to strongly affects the absorption capability of the atmosphere over Rome." Please explain more clearly the connection between the PM macro-components and the absorption capability of the atmosphere over Rome. From the Results I mostly understand from page 10, lines 285-286, that "Scatter plot of SSA400 versus the SOIL component (Figure4) shows a slight negative correlation (R= -0.54), whereas no other correlation is visible for the other components and other optical and physical parameters. " In case it is related to the theoretical calculations of page 8, lines 260-265, please explain in more details in Methodology how you have used the model.

The sentence in the abstract about the impact of the main macro-sources concentration on the absorption capability of the atmosphere over Rome was inferred from the theoretical calculations performed by the Rstar model, and not from the results related to the analysis of the ground based measurements. This is because, as also written in the text, columnar absorption properties and in situ measurements may not provide correlated information. Therefore, we agree with the observation of the Reviewer and we changed the sentence in the abstract as: "The modulation of their concentration, according to theoretical calculations, is expected to strongly affect the absorption capability of the atmosphere over Rome".

We also added the following paragrapgh in the Methodology section explain how the model has been qualitatively used: " Finally to help the understanding of the possible different effects of PM10 macro-components concentration on the atmosphere over Rome, the imaginary parts of refractive index of each fundamental materials in the Rstar model, were taken as reference. Rstar is a radiative transfer model (Nakajima and Tanaka 1986) able to simulate the radiation fields in the atmosphere-land-ocean system at the wavelength range 0.17 – 1000 ⸏m. Eight fundamental materials (water, dust-like, sea salt, volcanic ash, yellow sans, ice, water-soluble, soot and 75%H2SO4) are considered to assemble a three component internal mixture for each of the ten particles model types (Water, dust-like, volcanic-ash, rural, urban, yellow sand, ice, soot, 75%H2SO4, sea spray, tropo). In this study the refractive indexes for sea salt, soot and dust-like fundamental materials were taken as reference".

Add to the methodology the use of Lidar for detection of Sahara dust events.
The following sentence has been added in the Metodology section: "During the same campaign, the presence of Saharan dust over Rome was detected by manually inspecting the Lidar Backscatter ratio at 532 nm. Days showing aerosol above the Boundary layer, and the simultaneously check of the Hysplit (Draxler et al., 1998) back-trajectories (bringing airmass from Saharah reagions), were classified as "dusty"".

Minor comments:
Abstract Page 1 L28: "...is the primary parameter affecting the surface irradiance. . ..". Please specify that it is for clear sky at Rome. In some other sites, total ozone can the the primary parameter for clear skies. And if not clear skies, then cloudiness has an important role.

DONE
Page 1, line 38. Can you specify why the aerosol influence on UV is still uncertain.
The following sentence has been added "because in this wavelength region the columnar absorbing and scattering properties of suspended particles are not deeply inspected as in the visible spectral range".

Page 4, lines 108-111, Why is this mentioned here, if UVI used in the study is calculated using SHICRIVM? I suggest to remove those line.

The sentence has been reformulated as follow: "The SHICrivm algorithm, used to obtain the biologically effective UV irradiance, as explained in the section 3, compensates for the missing contribution of wavelengths longer than 325 nm. Based on considerations for similar corrections in the Brewer operating software (Fioletov et al., 2004), we estimate an uncertainty <2% in the UV index value for solar zenith angles <70° due to this extrapolation."

Page 5, lines 142-148, for people who are not familiar with cloud screening of aerosol measurements, the explanation is not clear. Please reformulate the reason for rejecting SSAs lower than 0.70.

The sentence has been reformulated as follow: "The simulation of an atmosphere contaminated by both dust-like and water insoluble aerosols brought to SSA400 values of about 0.70. Simultaneously values varying between 0.71-0.75 were retrieved testing a cirrus contamination case by enhancing the coarse mode for simulating the presence of ice particle types (cirrus particles model of the World Climate Programme report, Deepak and Gerber, 1983). Following these results, SSA400 values lower than 0.70 were rejected in this study because considered unrealistic, but it should be taken into account that values between 0.71 and 0.75 could contain information on both dust presence and cirrus-cloud contamination."

Page 5, line 158, Which are the highest UVI values (give some numbers).
The following sentence has been added "(a peak of 12.3 at Plateau Rosà, 3500 m a.s.l., in Valle d'Aosta Region, Italy)"

Page 5, line 159, How the clear sky spectra were selected?
They are selected according to : Alexandrov, M. D., A. Marshak, B. Cairns, A. A.Lacis, and B. E. Carlson (2004), Automated cloud screeningalgorithm for MFRSR data, Geophys. Res. Lett., 31, L04118,doi:10.1029/2003GL019105
The reference has been added in the text

Page 7, lines 224-225. It would be good to show the AERONET data also for the other years. Then it would be easier to the agreement/disagreement between the two instruments.
The comparison for the entire period has been shown, Figure 1 updated, and a Reference (Di Ianni et al., 2018) has been added.

Page 7, lines 225-226, Please move the AERONET explanation into the Section Methodology. Including the use of different wavelengths than for POM.
We retain that the use of AERONET retrievals in a different location than the one under study, is not part of the methodologies used in this study. They are merely used to show that the less famous SKYNET products are in good agreement with the AERONET ones therefore, we prefer leave the very short description of AERONET inversion and difference in wavelengths, in the results section.

Page 7, line 230, How did you defined that agreement is significant?
The following sentence has been written: "the agreement between the AERONET and SKYNET properties is mostly within the SKYNET standard deviations".

Page 7, line 235, Monthly averages of the total ozone content values and . . ..
The highest Ozone monthly measured values has been added in the text

Page 8, line 236, Please explain which kind of seasonal variability (higher in spring. . .) and give the highest values.

The following sentence has been written: "The seasonal ozone behavior is typical of mid-latitude sites, with highest values measured in spring and particularly in April 2010 (385 D.U. )  and March 2016 (374 D.U.).".

Page 8, line 238, I don't see the cumulated precipitation and pressure in the Figure.

Following the suggestion of another reviewer, they have been deleted but we forgot to delete the sentence too. Thanks.

Page 8, line 239, only the SSA400 and UVI is plotted in Figure 2. Not the AOD or Ang.

Corrected

Page 8, line 245, I don't understand the explanation, as aren't the UVI measurements performed under clear skies?

As said in the Methodology : "clear sky conditions (no clouds over the sun) were used to retrieve UV index values". This doesn't exclude the presence of scattered clouds in the sky, but can alter the UVI value. In particular, we may expect that a higher occurrence of scattered clouds conditions, corresponding to lower UVI values passing the cloud screening procedure (no cloud over the sun) , may be associated with periods with high precipitation during short-lived weather spring-summer disturbances.
We slightly changed in this sense the explanation.

Page 9, Figure 1 caption, Explain the red point also in the Figure caption. Add "monthly means" and for UVI at local noon.

Red points explained. "Monthly averages" is used in the text instead of "mean" to avoid the repetition, few words later, with "Annual means". For homogeneity is better leaving "averages" also in the caption. Added at LOCAL NOON for UVI

Page 10, Figure 2, local noon UVI under clear sky?

"Local noon"  has been added. Clear sky has not been added, because it is explained in the methodology that only cloud screened values are used ( no clouds over the sun)

Page 12, lines 314 and 315: Why not to try to analyse the aerosol absorption optical depth AAOD (1-SSA)*AOD?

As shown in Fig 3, a clear dependence of AOD on UVI is visible, but not on SSA. Therefore, the dependence of AAOD on UVI is expected to be mostly caused by the AOD dependence rather than the SSA one. Nevertheless, we tried analyzing the Absorption Angstrom exponent (AAE), whose information would have been very interesting. Unfortunately the Skyrad 4.2 pack, used in this study, has no smoothness spectral constraints on refractive indexes, as conversely AERONET and Skyrad-A have (Kudo et al., 2016 doi:10.5194/amt-9-3223-2016). This means that SSA values at 400 nm could be good  (as demonstrated by the comparison with AERONET in Figure 1), but the SSA spectral dependence,  at the basis of the AAE calculation is not.

Page 12, lines 316-319: What was the criteria for the chosen values to be used to make the division into subgroups?

The values separating the different groups were determined according to the frequency distributions of the two variables for the entire investigation period, shown in Figure 7. This is already stated in the text.

Page 12, line 337: How did you calculate the corresponding change of UVI* to be about 2?

It is from Figure 6, plot of UVI* vs Ang400-500.It has been added in the text

Page 13, line 345: Can you give any reasons why your results differ from those of Anton et al. 2011?

An explanation has been given at the end of section 4.

Page 13, line 354: There is a missing verb in the sentence + add at 30 degree, if so.
corrected

Figures 8 and 9, Move the heading of the color panels on the top of the color panel.
We are sorry but this is not possible because the headings are long and the center of the titles of each plot will be compromised.

Page 16, line 398, Where did you show results of analysis using AOD at 500 nm?
AODs at 400 and 500 nm were used to calculate Ang400-500. It has been explained in the text.

Page 16, line 401-403, Here again, why not to study the AAOD?
Please see the answer to your question above

Page 16, lines 413-415. What supports your assumption that the five macro – sources have not changed in the last years?
Generally, stable conditions exist during summer seasons in Rome characterized by a constant contribution of sea breeze during daytime. The SOIL source represents the most consistent contribution to the PM mass because the more aridity of soil during summer period lead to a higher resuspension of crustal-origin components operated by wind and vehicular traffic (Perrino et al., 2016, INDOOR AIR doi:10.1111/ina.12235). The stability of this situation supports the assumption we did. Many studies have been performed on the chemical analysis of the PM composition, but always related to short periods. In fact, it is impossible studying the PM chemical composition for a period long as the one considered in this study.

And as I wrote in the General Comment, I don't understand what supports the sentence "the variations in the absorption capability of the atmosphere over Rome were attributed to the different absorption characteristics of the macro-components and their modulation of concentration in the atmospheric mixture."
Please see the answer to your question at the beginning of the file.

Technical corrections:
In general the text contains long sentences, which are difficult to follow. Paragraph breaks are missing, e.g. in the section Results.
Paper has been re-read and corrected for long sentences.

Page 5, New paragraph cut between lines 152 and 153.
Done
Same for line 158 before Spectral UV...
Done

---

## Author Comment (AC3) · 10 Jan 2020

**Reviewer #1**

The authors present their findings on the impact of aerosol optical properties on the UV Index.

a) Main comment:

The title promises results in respect to the UV Index. Therefore it would be valuable to find the most important findings in respect to the UV Index already in the abstract. (PS: it is not really interesting where the Brewer is mounted.)

The abstract has been focused more on the findings about aerosol effect on UVI: "The aerosol optical characteristics in the urban area of Rome were retrieved over a period of 7 years from March to September 2010-2016. The impact of aerosol single scattering albedo (SSA), optical depth (AOD), estimated at 400 nm, and Ångström exponent on the ultraviolet (UV) index has been analyzed. Aerosol optical properties are provided by a PREDE-POM sun-sky radiometer of the ESR/SKYNET network and the UV index values were retrieved by a Brewer spectrophotometer both located in Rome. Chemical characterization of urban PM10 (particulate matter 10 micrometers or less in diameter) samples, collected during the URBan Sustainability Related to Observed and Monitored Aerosol (URBS ROMA) intensive filed campaign held in summer 2011 in the same site, was performed. PM macro-components were grouped in order to evaluate the contribution of the main macro-sources (SOIL, SEA, SECONDARY INORGANIC, ORGANICS and TRAFFIC). Their contributions were assumed not substantially changed in the other years under study, due to the general stable conditions during summer seasons in Rome, as reported by the literature. The modulation of their concentration, according to theoretical calculations, is expected to strongly affect the absorption capability of the atmosphere over Rome. The surface forcing efficiency, provided by the decreasing trend of UV index with AOD, which is the primary parameter affecting the surface irradiance during clear sky conditions in Rome, was found very significant, probably masking the dependence of UV index on SSA and Ångström exponents. Moreover it was found greater for larger particles and with a more pronounced slope at the smaller solar zenith angle. In Rome large particles are generally less absorbing since related to the presence of SOIL and SEA components in the atmosphere. The former contribution was found much higher in summer months because of the numerous episodes of Saharan dust transport".

Also the "conclusions" should focus on influence of aerosols on UV Index.

The conclusion were corrected removing the details about the average values of the aerosol optical properties and the percentage of PM10 components retrieved during URBS ROMA, both already described in the result section.

b) Minor comments:

l.16-17: 2010-2016 is it 6 or 7 years?

7, corrected

l.20: Optical data (abstract and introduction): please provide details;

Replaced with aerosol optical properties

l.22: delete

Deleted.

l.23: PM10 (abstract and elsewhere): provide definition

Done

l.30: SOIL and SEA type aerosols?

They are clusters in which the mixture of PM macro-components was divided to estimate the contribution of the main macro-sources. Their each composition is explained at the end of section 3. It has been now explained in the abstract.

l.82+l.122: … and at several scattering angles in the almucantar geometry: please clarify, what is the difference/meaning, otherwise delete

Corrected

l.123: … official code … is it computer code?

Corrected

l.147: Cost-713 not found in references

It already is in the list of references, after C.I.E. A separating line was missed. It as been added

l.163: … ozone air mass….. Please explain. I think that this a technical term is known in the Brewer/Dobson community but rather unknown for other readers.

Explained

Figures: please check labelling of all figures: e.g. Fig.1 axis (second last panel),

Corrected

labelling of color scales,….

Checked

---

## Editor Decision (ED1)

Dear authors,

I have read the reviewer comments and the answers.

To my point of view the paper still has a number of major methodological issues and also issues on describing the assumptions made in this work in order to end up in the conclusions.

**Major aspects**

The work has a major disadvantage. It tries to describe the effect of aerosol properties in the UV Index not having any aerosol properties measured at this region. So in order to end to the current conclusions there are the following assumptions:

- Ang. Exp derived from 400-500 range is accurately describing AOD at 305-315nm which is the effective (or the most important) wavelength range for UVI.

This can not be true as the introduced uncertainty is aerosol type dependent

- SSA in the visible range is equal or proportional with the one at UVB range for all aerosol types.

It is mentioned in the introduction that SSA spectral dependence is depending on the aerosol type.

 PM10 analysis ("In fact, assuming that the in situ measurements are representative of the entire column,..."). Based on the text and fig. 3 this is difficult to assume as there are a number of dust cases where is commonly known that aerosols can be found a lot heigher than the surface.

These three issues have to be re-discussed and relevant uncertainties and discussion has to be included.

Authors correlated UVI with AExp, AOD and SSA separately. Maybe it is a way to face the difficulties risen from the previous mentioned comment.

But in general. If SSA measurements are available UV changes are proportional to

AOD\* (1-SSA). However, the spectral dependence of SSA is obviously affecting the results here. An example:

Figure 9a slope = -1.77 and Figure 9b slope = -2.96. That shows that for a unit of aerosol optical depth the decrease of UV index in an absorbing (at visible range) environment (fig. 9a SSA<0.85) is less than the one with less absorption (fig 9b SSA>0.85). Even combined with the intercepts fig 9a reports a ~25% reduction of UVI per unit of AOD and fig 9b a ~36% reduction per unit of AOD.

Is this possible ?

Possibly it means SSA spectral dependence affects this analysis.

And this SSA spectral dependence also probably linked with AOD (through different aerosol types). Something that also Bais et al., 2005 (Effects of aerosol optical depth and single scattering albedo on surface UV irradiance". Atmospheric Environment, 39, 1093-1102, 2005) has shown.

Same is valid for figures 9c and 9d. Still lower SSA cases (9c) are linked with smaller UV changes for the same (a unit) of AOD and also theoretically larger air masses ( $\theta$ =40) should be linked with higher UV changes for the same AOD and SSA due the increased path of the atmosphere where the UV attenuates due to aerosols.

**Other comments**

**Abstract**

"The surface forcing efficiency, provided by the decreasing trend of UV index with AOD, which is the primary parameter affecting the surface irradiance during clear sky conditions in Rome, was found very significant, probably masking the dependence of UV index on SSA and Ångström exponents."

In general, to quantify the effect of AOD, and SSA separately you need to keep one of the constant especially here that they are interconnected. Theoretically the effect of AExp in UV here is just the effect of extrapolating correctly from the visible to the UV range.

"Moreover it was found greater for larger particles and with a more pronounced slope at the smaller solar zenith angle." I can not understand this sentence.

**Introduction**

"because in this wavelength region the columnar absorbing and scattering properties of suspended particles are not deeply inspected as in the visible spectral range."

I would suggest "because aerosol absorption properties in the UV are more difficult to be determined compared with the visible range"

(SSA), that change to (SSA) that

Optical depth (AOD) -> aerosol optical depth (AOD)

Aerosol optical depth = AOD , single scattering albedo = SSA from then on to the whole document.

Especially in winter - (I think in all seasons)

"di Sarra et al. (2002), Panicker et al. (2009), and Antón et al. (2011), among others, have shown that an increase of AOD induces a reduction of the UV index (UVI), an effective parameter to quantify the potentially harmful effects of UV radiation."

I do not understand this paragraph. Increase of AOD will lead to a UV decrease this is trivial. But how much it depends on other parameters and also by the use of AOD at UV wavelengths and not in 400nm. An effective aerosol related parameter not related with AOD but more with SSA and other aerosol optical properties can be defined as the aerosol radiative forcing efficiency (RFE) (e.g. see Kazadzis et al, 2009 (www.ann-geophys.net/27/2515/2009/). There is also a report there on how SSA can affect the RFE in an environment with much similarities as Rome.

Aerosol PREDE/POM measurements. You need to describe the aerosol properties you use in this study (AOD, SSA, Ang. Exponent) of which wavelengths and what is the uncertainty of these measurements.

"For sza>40, as in wintertime", I think For sxa>40 is enough as straylight and cosine effects mentioned here are only related with SZA and not seasons.

The performance of the Brewer instrument for UV measurements was controlled every two years till 2014 through intercomparisons to the traveling reference QASUME UV spectroradiometer (Groebner et al., Applied Optics, 44 (25) 2005).

I would also propose to put this paragraph starting "the performance of the Brewer ...till ... Sianni et al., 2013)" in the end of this section after ..extrapolation"

SHICRIVM algorithm needs a reference

The elastic LIDAR ... days affected by dust". How ? (reference or text).

**Methodology**

Infact - > In fact

"To point out the possible effect of aerosol optical characteristics measured at 400 nm on UVI\*, AOD400, SSA400, Ang and Ang400-500 were analyzed as function of UVI\* at the two fixed solar zenith angles, taking estimations of aerosol parameters and UVI\* within ±5 minutes."

As said this is my main concern for this paper. The representativeness of aerosol properties in the UV solely by measurements in the visible.

AERONET and Skynet comparison: I think this paragraph is confusing.

On the one hand when results agree, authors conclude that results are within the Skynet standard deviations (this also should be replaced by Skynet uncertainty), but for March and May the authors refer to spatial issues due to the non collocation of the instruments.

"In fact, assuming that the in situ measurements are representative of the entire column,..."

How can this be possible when there are a number of dust events (fig 3) that in general affect much more the columnar properties due to the presence of aerosol plumes higher in the atmosphere ?

"The general behavior of observed five micro sources.. has been assumed not substantially changed in the last years".

This is difficult to assume looking at the SSA variability for the 7 year period on fig. 1 and the text: "SSA400 vary between a minimum value of 0.84±0.08 (observed in 2016) and a maximum of 0.97±0.03 (observed in 2015)." This is a huge change in absorption that for sure has to do with changes in the aerosol type composition in the atmosphere.

**Conclusions**

Need to follow a number of previous aspects mentioned here.

---

## Author Response (AR2)

Dear authors,

I have read the reviewer comments and the answers. To my point of view the paper still has a number of major methodological issues and also issues on describing the assumptions made in this work in order to end up in the conclusions.

Dear Editor, we submit a new version of this study answering to many of your questions. Firstly, we implemented the dataset including measurements at 340 nm and extending the period under study up to 2020 and no more 2016. Secondly, we re-analyzed the dataset with a new code: Skyrad_MRIv2. pack (Kudo et al., 2021) to retrieve the aerosol optical properties from the sun-sky radiometer measurements. The use of a new computer code, more accurate in retrieving the wavelength dependence of SSA, allowed us to obtain better results.  Finally, the structure has been slightly modified with the following sections: 1 Introduction; 2 data; 2.1Measurements site; 2.2 Sun-sky radiometer measurements; 2.3 Brewer measurements; 2.4 Particulate Matter samples collected at the surface. 3 Methodology; 3.1 Sun-sky radiometer retrieval method; 3.2 Brewer retrieval method; 3.3 PM samples chemical analysis; 3.4 Optical properties of surface aerosol from a radiative transfer model; 3.5 Assessment of the dependence of UVI on the aerosol optical properties; 4 Results; Validation of the method to extrapolate the aerosol properties to 340 nm; 4.2 UVI dependence on aerosol optical parameters; 5 Conclusions.

Major aspects

The work has a major disadvantage. It tries to describe the effect of aerosol properties in the UV Index not having any aerosol properties measured at this region.

We implemented the dataset including measurements at 340 nm and extending the period under study.We re-analyzed the dataset with a new code: Skyrad_MRIv2. pack (Kudo et al., 2021) to retrieve the aerosol optical properties from the sun-sky radiometer measurements. In order to relate the UVI to the aerosol optical properties, these latter were determined at the 340 nm. Measurements at this wavelength were started in 2016 ("dataset 2"), while the shortest wavelength was 400 nm prior to that date ("dataset 1"). Hence, to increase the length of the aerosol dataset at the shortest measured wavelength and cover a larger overlapping period with the UVI series from the Brewer, we developed a new physically-based method to extrapolate the aerosol optical depth and aerosol properties from longer wavelengths (400 nm and above) down to 340 nm for both dataset 1 and 2, using the Aerosol Optical Properties (AOP) program included in the Skyrad MRIv2 package. Then, to assess the accuracy of the method, we compared the

outcome of this extrapolation with the retrieval obtained using all available wavelengths, including 340 nm (period dataset 2). Based on the very good results of such a comparison (Sect. 4.1), we always used the extrapolated data in the entire analyzed period for consistency. Replacing the retrievals based on observations at 340 nm with the extrapolation is therefore not expected to affect the findings of this study.

So in order to end to the current conclusions there are the following assumptions:

- Ang. Exp derived from 400-500 range is accurately describing AOD at 305-315nm which is the effective (or the most important) wavelength range for UVI. This can not be true as the introduced uncertainty is aerosol type dependent

We now used Angstrom exponent calculated between 340 and 500 nm

- SSA in the visible range is equal or proportional with the one at UVB range for all aerosol types. It is mentioned in the introduction that SSA spectral dependence is depending on the aerosol type

We now used SSA calculated at 340 nm. The use aerosol optical properties determined at a "measured" wavelength as close as possible to the one corresponding to the maximum of the erythemally-weighted solar spectrum (usually <320 nm, depending on the solar zenith angle) is necessary, in order to have the possibility to validate the retrievals as shown in the new section 4.1.

- PM10 analysis ("In fact, assuming that the in situ measurements are representative of the entire column,..."). Based on the text and fig. 3 this is difficult to assume as there are a number of dust cases where is commonly known that aerosols can be found a lot heigher than the surface.
The sentence has been removed and the use of the results from URBS campaign has been reviewed.

These three issues have to be re-discussed and relevant uncertainties and discussion has to be included.
The expected uncertainty in the retrieval products at near-ultraviolet, visible, and near-infrared wavelengths is less than 0.04 for AOD, and less than 0.05 for SSA, as discussed in the paper Kudo et al., 2021).

Authors correlated UVI with AExp, AOD and SSA separately. Maybe it is a way to face the difficulties risen from the previous mentioned comment. But in general. If SSA measurements are available UV changes are proportional to  AOD* (1-SSA). However, the spectral dependence of SSA is obviously affecting the results here.
We estimated $AAOD_{340}$ and its dependence on UVI is now shown in the text

An example:

Figure 9a slope = -1.77 and Figure 9b slope = -2.96. That shows that for a unit of aerosol optical depth the decrease of UV index in an absorbing (at visible range) environment (fig. 9a SSA<0.85) is less than the one with less absorption (fig 9b SSA>0.85). Even combined with the intercepts fig 9a reports a ~25% reduction of UVI per unit of AOD and fig 9b a ~36% reduction per unit of AOD.

Is this possible ? Possibly it means SSA spectral dependence affects this analysis. And this SSA spectral dependence also probably linked with AOD (through different aerosol types). Something that also Bais et al., 2005 (Effects of aerosol optical depth and single scattering albedo on surface UV irradiance". Atmospheric Environment, 39, 1093-1102, 2005) has shown. Same is valid for figures 9c and 9d. Still lower SSA cases (9c) are linked with smaller UV changes for the same (a unit) of AOD and also theoretically larger air masses ( θ=40) should be linked with higher UV changes for the same AOD and SSA due the increased path of the atmosphere where the UV attenuates due to aerosols. The use of a new computer code, Skyrad_MRIv2, more accurate in the retrieving the wavelength dependence of SSA allowed us to obtain better results.

Other comments

Abstract

"The surface forcing efficiency, provided by the decreasing trend of UV index with AOD, which is the primary parameter affecting the surface irradiance during clear sky conditions in Rome, was found very significant, probably masking the dependence of UV index on SSA and Ångström exponents."

In general, to quantify the effect of AOD, and SSA separately you need to keep one of the constant especially here that they are interconnected. The use of the new computer code, Skyrad_MRIv2, allowed us to obtain more accurate results.

Theoretically the effect of AExp in UV here is just the effect of extrapolating correctly from the visible to the UV range. We now used Angstrom exponent calculated between 340 and 500 nm

"Moreover it was found greater for larger particles and with a more pronounced slope at the smaller solar zenith angle." I can not understand this sentence.

The description of the results has been strongly modified thanks to the new processing code and the availability of the new shorter wavelength measurements. The surface forcing efficiency showed that AOD is the primary parameter affecting the surface irradiance under clear sky conditions in Rome. SSA and the Ångström exponent are also identified as secondary influencing factors, i.e., the surface forcing efficiency is found to be greater for smaller zenith angles and for larger and more absorbing particles in the UV range (such as, e.g., mineral dust).

Introduction

"because in this wavelength region the columnar absorbing and scattering properties of suspended particles are not deeply inspected as in the visible spectral range."

I would suggest "because aerosol absorption properties in the UV are more difficult to be determined compared with the visible range"
The sentence has been changed, keeping the suggested meaning but with different words.

(SSA), that change to (SSA) that Optical depth (AOD) -> aerosol optical depth (AOD)
Aerosol optical depth = AOD , single scattering albedo = SSA from then on to the whole document.
We are sorry but we didn't understand this comment

Especially in winter - ( I think in all seasons)

"di Sarra et al. (2002), Panicker et al. (2009), and Antón et al. (2011), among others, have shown that an increase of AOD induces a reduction of the UV index (UVI), an effective parameter to quantify the potentially harmful effects of UV radiation."

I do not understand this paragraph. Increase of AOD will lead to a UV decrease this is trivial. But how much it depends on other parameters and also by the use of AOD at UV wavelengths and not in 400nm. The use aerosol optical properties determined at a "measured" wavelength as close as possible to the one corresponding to the maximum of the erythemally-weighted solar spectrum (usually <320 nm, depending on the solar zenith angle) is necessary, in order to have the possibility to validate the the retrievals as shown in the new section 4.1. In our case the shortest is now changed to 340nm and the dependence of UVI on aerosol properties at this wavelength has been studied.

An effective aerosol related parameter not related with AOD but more with SSA and other aerosol

optical properties can be defined as the aerosol radiative forcing efficiency (RFE) (e.g. see Kazadzis et al, 2009 (www.ann- geophys.net/27/2515/2009/). There is also a report there on how SSA can affect the RFE in an environment with much similarities as Rome. Aerosol PREDE/POM measurements. You need to describe the aerosol properties you use in this study (AOD, SSA, Ang. Exponent) of which wavelengths and what is the uncertainty of these measurements.

The expected uncertainty in the retrieval products at near-ultraviolet, visible, and near-infrared wavelengths is less than 0.04 for AOD, and less than 0.05 for SSA, as discussed in the paper Kudo et al., 2021). This has been added in the text.

"For sza>40, as in wintertime", I think For sxa>40 is enough as straylight and cosine effects mentioned here are only related with SZA and not seasons.

"as in wintertime" was removed

The performance of the Brewer instrument for UV measurements was controlled every two years till 2014 through intercomparisons to the traveling reference QASUME UV spectroradiometer (Groebner et al., Applied Optics, 44 (25) 2005).  I would also propose to put this paragraph starting "the performance of the Brewer ...till ... Sianni et al., 2013)" in the end of this section after ..extrapolation"

The text is in a new structure

SHICRIVM algorithm needs a reference

The reference is added

The elastic LIDAR ... days affected by dust" . How ? (reference or text). Methodology

This part has been removed

"To point out the possible effect of aerosol optical characteristics measured at 400 nm on UVI*, AOD400, SSA400, Ang and Ang400-500 were analyzed as function of UVI* at the two fixed solar zenith angles, taking estimations of aerosol parameters and UVI* within ±5 minutes."

As said this is my main concern for this paper. The representativeness of aerosol properties in the UV solely by measurements in the visible.

Now we moved the analysis to the 340 nm.

AERONET and Skynet comparison: I think this paragraph is confusing. On the one hand when results

agree, authors conclude that results are within the Skynet standard deviations (this also should be replaced by Skynet uncertainty), but for March and May the authors refer to spatial issues due to the non collocation of the instruments

This part is no more in the new text

"In fact, assuming that the in situ measurements are representative of the entire column,..." How can this be possible when there are a number of dust events (fig 3) that in general affect much more the columnar properties due to the presence of aerosol plumes higher in the atmosphere ? "The general behavior of observed five micro sources.. has been assumed not substantially changed in the last years". This is difficult to assume looking at the SSA variability for the 7 year period on fig. 1 and the text: "SSA400 vary between a minimum value of 0.84±0.08 (observed in 2016) and a maximum of 0.97±0.03 (observed in 2015)." This is a huge change in absorption that for sure has to do with changes in the aerosol type composition in the atmosphere.

The use of the results from URBS campaign has been reviewed.

---

## Author Response (AR3)

We thank the editor for the agreement in our updated version of the paper.

I think that the manuscript has been substantially improved and can be published in AMT.

I have only 3 minor comments

Line 118-119 a reference could be used for qasume.
J. Gröbner, J. Schreder, S. Kazadzis, A. F. Bais, M. Blumthaler, P. Gorts, R. Tax, T. Koskela, G. Seckmeyer, A. R. Webb, "A travelling reference spectroradiometer for routine quality assurance of spectral solar ultraviolet irradiance measurements", Applied Optics, 44 (25) 2005

The reference has been added

Figure 2
SSA error bars are the standard deviation that it is lower compared to the reported 0.05 error on SSA retrieval. Maybe this can be mentioned in the figure caption.

The comment has been added.

UVI "effective" wavelengths are around 305-310nm depending on solar elevation, on the contrary aerosol properties are used at 340nm. Since AOD305 is theoretically higher than AOD340 for the same instant and the relationship between AAOD 305 and AAOD 340 depends on SSA spectral behavior in the UVB, maybe it should be mentioned that part of the analysis results (in figure 5 for example) and results of table I could be partly affected by the spectral behavior of both AOD and SSA in the 305-340 nm range.

Also this comment has been added